# Prevention and management of type 2 diabetes mellitus in Uganda and South Africa: Findings from the SMART2D pragmatic implementation trial

David Guwatudde[1]*, Peter Delobelle[2,3,4], Pilvikki Absetz[5,6], Josefien Olmen Van[7,8], Roy William Mayega[1], Francis Xavier Kasujja[1], Jeroen De Man[7], Mariam Hassen[3], Elizabeth Ekirapa Kiracho[9], Juliet Kiguli[10], Thandi Puoane[3], Claes-Goran Ostenson[11], Stefan Peterson[9,12,13,14], Meena Daivadanam[13,14,15], SMART2D Consortium[¶]

1 Department of Epidemiology and Biostatistics, School of Public Health, Makerere University College of Health Sciences, Kampala, Uganda, 2 Chronic Disease Initiative for Africa, University of Cape Town, Cape Town, South Africa, 3 School of Public Health, University of the Western Cape, Cape Town, South Africa, 4 Department of Public Health, Vrije Universiteit Brussel, Brussels, Belgium, 5 Collaborative Care Systems Finland, Helsinki, Finland, 6 Faculty of Social Sciences, Tampere University, Tampere, Finland, 7 Department of Family Medicine and Population Health, University of Antwerp, Antwerp, Belgium, 8 Department of Public Health, Institute of Tropical Medicine, Antwerp, Belgium, 9 Department of Health Policy, Planning and Management, School of Public Health, Makerere University College of Health Sciences, Kampala, Uganda, 10 Department of Community Health and Behavioral Sciences, School of Public Health, Makerere University College of Health Sciences, Kampala, Uganda, 11 Department of Molecular Medicine & Surgery, Diabetes & Endocrine Unit, Karolinska Institutet, Stockholm, Sweden, 12 International Maternal and Child Health Unit, Uppsala University, Uppsala, Sweden, 13 Department of Women's and Children's Health, Uppsala University, Uppsala, Sweden, 14 Department of Global Public Health, Karolinska Institutet, Stockholm, Sweden, 15 International Child Health & Nutrition Research Group, Uppsala University, Uppsala, Sweden

¶ Membership of the SMART2D Consortium is provided in the Acknowledgments.
* dguwatudde@musph.ac.ug

**Data Availability Statement:** A fully de-identified data set is provided as S1 Data.

## Abstract

Health systems in many low- and middle-income countries are struggling to manage type 2 diabetes (T2D). Management of glycaemia via well-organized care can reduce T2D incidence, and associated morbidity and mortality. The primary aim of this study was to evaluate the effectiveness of facility plus community care interventions (integrated care), compared to facility only care interventions (facility care) towards improvement of T2D outcomes in Uganda and South Africa. A pragmatic cluster randomized trial design was used to compare outcomes among participants with T2D and those at high risk. The trial had two study arms; the integrated care arm, and the facility care arm; and in Uganda only, an additional usual care arm. Participants were enrolled at nine primary health facilities in Uganda, and two in South Africa. Participants were adults aged 30 to 75 years, and followed for up to 12 months. Primary outcomes were glycaemic control among participants with T2D, and reduction in HbA1c > = 3 mmol/mol among participants at high risk. Secondary outcomes were retention into care and incident T2D. Adjusted analysis revealed significantly higher retention into care comparing integrated care and facility care versus usual care in Uganda and integrated care versus facility care in South Africa. The effect was particularly high

**Funding:** This study was part of the SMART2D project funded by the European Commission's Horizon2020 Health Coordination Activities (Grant Agreement No 643692) under call "HCO-05-2014: Global Alliance for Chronic Diseases: prevention and treatment of type 2 diabetes" (DG, PD, PA, JOV, RWM, FXK, JDM, MH, EEK, JK, TP, CGO, SP, MD). The Uganda site was co-funded by the Swedish International Development Cooperation Agency (Sida) capacity-building grant to Makerere University 2015-2010, Project number HS 343 (DG, RWM, FXK, EEK, JK). The contents of this article are solely the responsibility of the authors and do not reflect the views of the funders of the SMART2D Project. The funders had no role in study design, data collection and analysis, decision to publish, or preparation of the manuscript.

**Competing interests:** The authors have declared that no competing interests exist.

among participants at high risk in Uganda with an incident rate ratio of 2.46 [1.33–4.53] for the facility care arm and 3.52 [2.13–5.80] for the integrated care arm. No improvement in glycaemic control or reduction in HbA1c was found in either country. However, considerable and unbalanced loss to follow-up compromised assessment of the intervention effect on HbA1c. Study interventions significantly improved retention into care, especially compared to usual care in Uganda. This highlights the need for adequate primary care for T2D and suggest a role for the community in T2D prevention.

**Trial registration number:** ISRCTN11913581.

## Introduction

The global burden of type 2 diabetes mellitus (T2D) and associated morbidity and mortality continues to rise, especially in low- and middle-income counties (LMICs) [1]. The 2019 global burden of disease estimated that diabetes among adults would increase from an estimated prevalence of 8.8% in 2015, to a prevalence of 10.4% by 2040; and most of the increase would occur in LMICs. The same study further showed that in 2019, diabetes ranked 8[th] as the leading cause of deaths and Disability Adjusted Life Years (DALYs) lost, up from 20[th] in 1990 [2]. Global estimates also show that sustained glycemic control among patients with diabetes is only 50% [3]. Data from STEPS surveys conducted in 40 LMICs showed that of all patients with T2D in these countries, only 38% are on treatment, and 23% achieve glycemic control [4]. Furthermore, data from the South African National Health and Nutrition Examination Survey (SANHANES-1 [2011–2012]) found that among individuals with T2D, 45.4% were unscreened, 14.7% were screened but undiagnosed, 2.3% were diagnosed but untreated, and 18.1% were treated but uncontrolled; suggesting that 80.5% of the population with diabetes had unmet need for care [5].

To manage the disease adequately, people living with T2D do not only need timely care but also diabetes self-management education and ongoing diabetes self-management support [6]. Similarly, to prevent or delay onset of T2D, people at high risk should receive preventive care, education and support to manage risk for T2D. There is strong evidence demonstrating that in the absence of risk reduction interventions, approximately 5%– 10% of individuals with pre-diabetes progress to diabetes annually [7]. Several randomized controlled trials conducted in high income countries (HICs), including the Diabetes Prevention Program (DPP) [8], and the Finnish Diabetes Prevention Study (DPS) [9], also demonstrate that lifestyle/ behavioral interventions are highly effective in preventing T2D.

Given the increasing burden of diabetes in LMICs, and the lack of targeted prevention strategies coupled with inadequate care for T2D, there is an urgent need to identify context relevant interventions with the potential to work across different income settings, aimed at preventing progress to diabetes among individuals at high risk, as well as improving diabetes management outcomes among patients with diabetes.

The 'Self-Management Approach and Reciprocal learning for Type 2 Diabetes' (SMART2D) was a collaborative project implemented between January 2015 and December 2019 in two different settings: a rural area in Uganda (a low-income country), and a semi-urban township in Cape Town, South Africa (a middle-income country). The main objective of the project was to formulate and implement a contextually appropriate self-management strategy for the prevention and control of T2D in each setting and to evaluate its outcomes [10, 11]. As part of the project, a pragmatic cluster randomized trial was designed and implemented to evaluate the effectiveness and implementation of integrated facility plus community

care interventions, compared to facility care only, in improving T2D management outcomes among patients with diabetes, and individuals at high risk of T2D in Uganda and South Africa. The Self-determination theory (SDT) [12] was used as part of the theoretical framework for our study and its relevance was tested using the baseline data of the SMART2D trial. The aim of the analysis reported in this article was to evaluate the effectiveness of the interventions towards improvement of T2D prevention and management outcomes in the two countries.

## Materials and methods

### Ethics statement

In Uganda, approval for the study was obtained from the Research and Ethics Committee of the Makerere University School of Public Health (reference number 426), and from the Uganda National Council for Science and Technology (reference number HS 2118). In South Africa, approval was obtained from the Biomedical Science Research Ethics Committee of the University of the Western Cape (reference number BM/17/1/36). Written informed consent was obtained from eligible subjects before enrollment in the study. Subjects not enrolled in the study but had a fasting plasma glucose test reading of at least 6.1 mmol/L, and were not into care were advised to as soon as possible report to the nearest government owned health facility for further evaluation.

### Study design and setting

The study used a pragmatic cluster randomized trial design, to allocate clusters to study arms. Clusters were primary health care facilities and their respective catchment areas. In Uganda, a health facility was eligible for inclusion into the trial if it provided primary health care, and was at least 5 kilometers away from an already selected one. Using these two criteria, nine health primary health care facilities were selected with the aim of being able to randomize three facilities to each of the three study arms. In South Africa, facilities were assessed for eligibility based on facilities being managed by the same governance structure and existence of a community-based platform for T2D services support. Based on these criteria, two primary health care facilities were selected. The primary study units were patients with diabetes, and individuals at high risk of T2D recruited within each of the study clusters. Recruitment of potential participants was then conducted within the catchment areas, and at the health facilities as described below.

In South Africa, the trial included two study arms; a facility plus community care (integrated care) arm, and a facility only care (facility care) arm. Because in South Africa facility care provides the required standard of care for patients with diabetes, the facility care arm served as the control arm with no study interventions; whereas the integrated care arm served as the intervention arm. In Uganda where primary care processes for diabetes are not regularly fully provided at primary health care facilities, the trial had three study arms including the two study arms as in South Africa, plus a usual care arm that served as the control arm. We made no interventions in the usual care arm except provision of diabetes medications to avoid stock outs.

### Eligibility criteria

Subjects were eligible for participation in the study if they fulfilled the following criteria: aged between 30 and 75 years, had resided in their respective communities for at least 6 months, had no plans of out-migrating from the study area over the next 12 months; able to provide written informed consent; allow home visits and follow-up contacts; and a diagnosis of diabetes of no longer than 12 months, or if classified as being at high risk of T2D. A subject was

classified as having diabetes if he/she had already been diagnosed with diabetes and was in care for diabetes at the study health facility, or if they had at least two fasting plasma glucose test readings of greater than 7.0 millimoles per liter (mmol/L) conducted by our study staff on two separate days within two weeks prior to enrollment into the study. In Uganda, a subject was classified as being at high risk of T2D if they had pre-diabetes, defined as having at least two fasting plasma glucose test readings of between 6.1–6.9 mmol/L conducted by our study staff on two separate days within two weeks prior to enrollment into the study. In South Africa, a subject was classified as being at high risk of T2D if they had a random plasma glucose test reading $\leq 11$ mmol/L, a BMI of at least 25, and having one or more of the following: hypertension, a cardiovascular disease, physical inactivity, family history of T2D, or previous gestational diabetes. Pregnancy and serious mental disability were exclusion criteria. Once eligibility was confirmed, written informed consent was obtained and the subject enrolled into the study.

## Interventions

The facility care arm comprised two intervention strategies including: 1) organization of care at facility level, and 2) strengthening of the patient role in self-management. The integrated care arm included the two intervention strategies in the facility care arm, plus three community intervention strategies including: 1) community mobilization (including dissemination of messages on healthy lifestyles to community members and key stakeholders); 2) creating a supportive environment by establishing peer support groups and identifying a care companion for each participant; and, 3) establishing a community extension link between the facility and the community for the care of patients. Interventions in this trial are hence related to clusters. In Uganda, study health facilities were supported with medication and diabetes diagnostic equipment, reagents and strips to address critical gaps in usual care that would otherwise influence trial implementation. Details of the various elements within each intervention strategy have been described elsewhere [13]. Participants were exposed to the intervention strategies for at least 9 months. Most of the intervention strategies were developed based on principles of the Self Determination Theory (SDT) [14]. The theoretical framework and development of the care interventions have been described elsewhere [11, 15].

## Measurements

Following enrollment, trained study staff administered a standardized questionnaire to obtain baseline data that included: socio-demographic characteristics (age, sex, level of education, marital status) and behavioral characteristics (alcohol use, tobacco use, self-reported levels of physical activity, food consumption patterns, foot care, and medical and medication history). Physical measurements included weight, height, waist circumference, blood pressure measurements; and a baseline glycated haemoglobin A1c (HbA1c).

Blood pressure (BP) was measured using a digital upper arm sphygmomanometer (Omron® M3 series, Omron Healthcare Inc.), with three measurements taken at least five minutes apart. The mean of the last two BP measurements was used in the analysis. Weight was measured using a digital weighing scale (Seca 813, Hanover, USA) and height was measured using a roll-up stadiometer (Seca 213, Hanover, USA). Body mass index (BMI) was calculated by dividing the participant's weight in kilograms by the height in meters squared (kg/$m^2$). Waist circumference was measured using a measuring tape with the participant in light clothing. Plasma glucose was measured using a point of care glucometer (Accucheck®) by Roche, which uses 10 microliters (μL) of capillary blood derived from a finger prick, and the Cobas b101 point-of-care analyzer (in Uganda) and the Alfinion AS100 Analyser (in South Africa) were used to measure HbA1c levels in millimoles/mole (mmol/mol) using 5 μL of

capillary blood to provide results that are free from hemoglobin variant interference, which is National Glycohemoglobin Standardization Program (NGSP) certified.

Each participant was followed for up to 12 months or until death, withdrawal of consent, or loss to follow-up. Follow-up of participants was completed at all country sites by the end of December 2019.

## Quality control of interventions administration

In both countries, a contextualized structured training program guided intervention administration including pre-intervention training as well as quality control during the implementation phase. In Uganda, regular support supervision visits were conducted to the health facilities to ensure that the interventions were implemented according to protocol and with sufficient dose of exposure, fidelity and reach. During the visits, study team members checked and re-enforced health worker compliance to key elements of the intervention e.g. ensuring no stock out of key medicines and testing equipment, compliance with treatment guidelines, task shifting, maintenance of the patient information system, and supporting the patient role in self-care. Similar support was given to the community level interventions. In South Africa, where the intervention focused mainly on the community interventions, quality assurance visits were implemented to supervise peer support groups to check for intervention fidelity; monthly mock peer group sessions were organized and refresher training in diabetes management. Record keeping practices were also supported to improve reporting feedback.

## Outcomes

The primary outcome among participants with diabetes was glycemic control at month 12, and among participants at high risk of T2D was reduction in HbA1c levels of at least 3 mmol/mol between baseline and month 12. Glycemic control was defined using the American Diabetes Association definition, i.e. having an HbA1c reading below 53 mmol/mol (7.0% using the Diabetes Control and Complications Trial method) [16] at month 12. Regarding reduction in HbA1c levels of at least 3 mmol/mol by month 12 among participants at high risk of T2D, we decided to use this cutoff based on findings from an intervention trial reported by Katula et al (2021) that aimed at reducing HbA1c levels among people with pre-diabetes [17]. In this study, investigators observed significant reduction in HbA1c levels, an average of -2.52 mmol/mol [95% CI = -2.89 –-2.16]. Based on this, we decided to use a cutoff reduction of 3 mmol/mol by month 12. Secondary outcomes included: i) retention into care defined as having returned for the month 12 clinic appointment and endline measurements done within 14 days of the appointment date, and, ii) incident T2D diabetes defined as a participant at high risk for T2D whose baseline HbA1c test reading was less than 48 mmol/mol (6.5%), but had an HbA1c test reading of 48 mmol/mol (6.5%) or greater at month 12.

## Sample size

A detailed description of sample size calculations has been reported elsewhere, showing the number of clusters with unequal sizes, the coefficient of intra-cluster correlation (ICC), and the other parameters used in the calculations [13].

## Data management and analysis

At each country site, research data was managed by trained data managers, with regular data cleaning, and quarterly upload into RedCap software onto a server via secure links. The database password was protected at all levels, with access only to authorized study staff.

Important to note is that although the units of allocation to study arms were the health facilities, the units of analysis were the individual participants. We report the percentage of participants with T2D that achieved glycemic control, the percentage of participants at high risk that achieved reduction in HbA1c of at least 3 mmol/mol between baseline and month 12, the percentage of participants at high risk that progressed to T2D by month 12 (incident T2D), and the percentage of participants with T2D and those at high risk that were retained into care. For the two primary outcomes (glycaemic control and reduction in HbA1c of at least 3 mmol/mol), participants that were lost to follow-up, their missing data were imputed using multiple imputations based on a "missing at random assumption". Imputations were created based on a fully conditional specification which imputes multivariate missing data on a variable-by-variable basis in an iterative fashion [18]. For the calculation of the imputation values, we used predictive mean matching, an implicit model which is assumed robust to misspecification. Standard Error (SE) estimates were pooled based on Rubin's rules across 10 imputed data sets. All model variables and additional extraneous variables (i.e. marital status, employment and income) were used as predictors. The procedure was done using the "Mice" package in R [19]. A sensitivity analysis was run in which missing outcomes were set equal to "not having the outcome of interest". For the secondary outcome retention into care, analyses were conducted using the intention-to-treat analysis strategy [20]. In this strategy, all participants enrolled in the trial are included in the analysis as recommended by CONSORT guidelines.

Adjusted analyses were conducted separately for each country. By design, within each country the data had a two-level structure, i.e. individual level (level 1) and health facility level (level 2). Multilevel modified Poisson regression modeling was used to take into account this hierarchical structure, with individual participants nested within the health facility of enrollment. The health facility indicator was included in the model as a random effect, except for South Africa where only one facility per arm was included. The modified Poisson regression model provides estimated incident rate ratios (IRR) of the outcome of interest, comparing rates of the outcome of interest across the different categories of independent variables. Modified Poisson regression modeling was preferred over logistic regression modeling to avoid underestimation of the standard errors for the estimated risk ratios that is usually encountered with logistic regression modeling when the prevalence of the outcome is greater than 10% [21, 22]. To adjust for imbalances of relevant population characteristics at baseline, adjusted analysis modeling was weighted using propensity scores. These scores were calculated using gradient boosted logistic regression, a supervised nonparametric machine learning technique available in R's 'twang' package [23], with the trial arm as the outcome and relevant baseline characteristics of participants as the covariates [24, 25].

Furthermore, we controlled for the various baseline covariates including socio-demographic characteristics (age, sex, education level, marital status), physical characteristics (BMI), behavioral characteristics (tobacco use, alcohol use, physical activity levels), and health status indicators (hypertension status and baseline HbA1c test reading). The covariates were predetermined during the planning of the study, and selection of these was based on theory, and previous similar research. All statistical analyses were performed using STATA version 14 [26] and R version 4.1.2 (2021-11-01) [27].

## Patient and public involvement

Local community leaders and other key stakeholder were involved in the design and development of the SMART2D intervention elements as described in detail elsewhere [10, 11]. At the end of the trial, the local community leaders, selected study participant representatives and care companions participated in a workshop of dissemination of findings from the trial to policy makers.

## Results

### Enrollment and follow-up

Between 11th January 2017 and 30th November 2018 health care facilities and individuals were screened for eligibility to participate in the study. In Uganda, out of the 101 facilities screened nine were selected, whereas in South Africa, out of the 9 facilities, two were selected. In Uganda, 28,175 subjects residing in the nine cluster areas were screened, out of which 801 were enrolled (424 with T2D and 377 at high risk). In South Africa, 1,584 subjects were screened at the selected health facilities out of which 566 were enrolled (281 with T2D and 285 at high risk). Figs 1 and 2 summarize the trial schema for Uganda and South Africa, respectively; giving details of the distribution of numbers from screening through enrollment, loss to follow-up, and number of participants analyzed by study arm; and Table 1 summarizes the number of participants by cluster.

### Baseline characteristics of participants

Details of the baseline characteristics of participants, stratified by country site and by study arm are presented in Table 2. Among participants with diabetes, significant differences between the study arms in Uganda were observed only in the percentage of participants with hypertension, which was significantly lower in the integrated care arm than the other two arms. In South Africa, participants in the integrated care arm had significantly lower mean age, BMI and percentage of participants with hypertension but higher mean HbA1c than participants in the facility care arm.

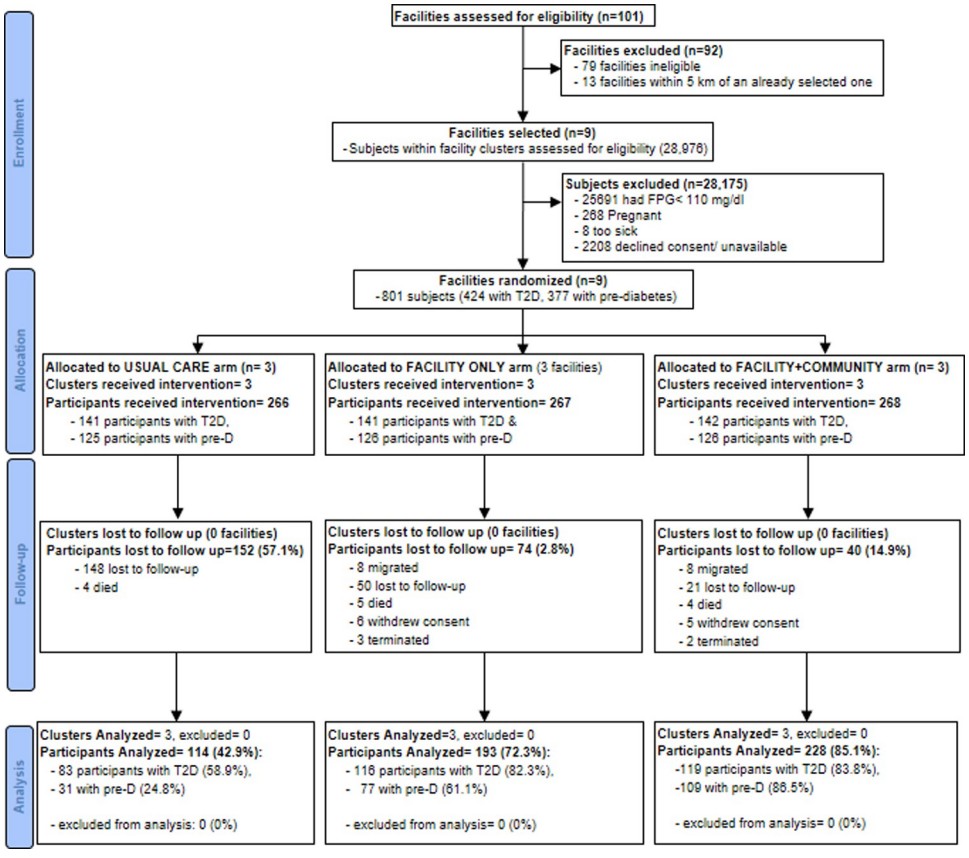

**Fig 1. Uganda CONSORT flow chart.**

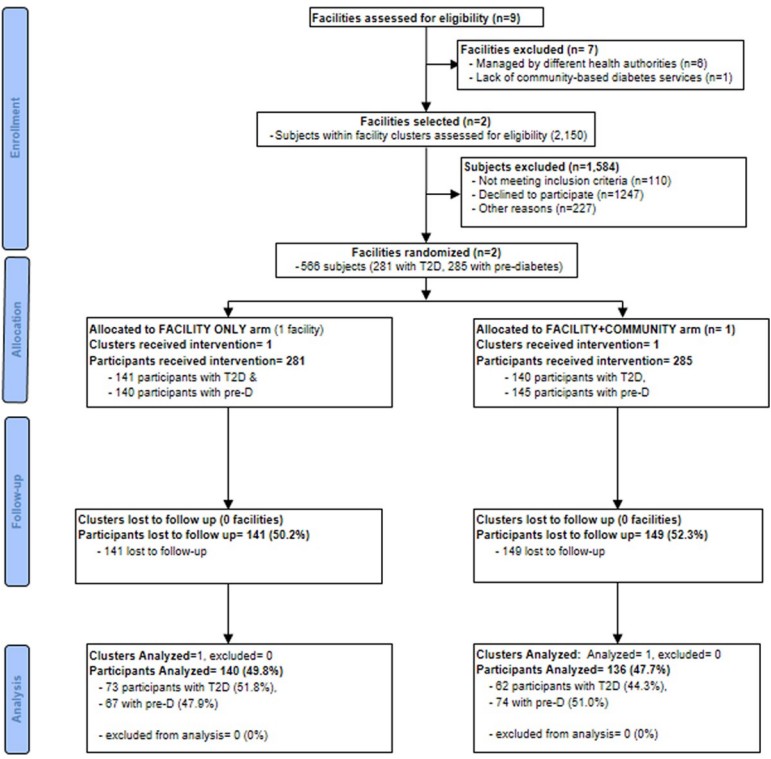

**Fig 2. South Africa CONSORT flow chart.**

Among participants at high risk for T2D in Uganda, significant differences were again observed between the study arms only in the percentage with hypertension, however in this group the usual care arm had the lowest percentage (Table 2). In South Africa, significant

**Table 1. Number of participants by cluster.**

| Trial arm | Name of Health Facility | Number of participants with T2D | Number of participants at high risk of T2D | Total |
|---|---|---:|---:|---:|
| **UGANDA** | | | | |
| Usual care | Lubira HC | 47 | 41 | 88 |
| | Namungalwe HC | 47 | 41 | 88 |
| | Kigandalo HC | 47 | 43 | 90 |
| | **sub-total** | **141** | **125** | **266** |
| Facility care | Busesa HC | 47 | 43 | 90 |
| | Kityerera HC | 47 | 42 | 89 |
| | Mayuge HC | 47 | 41 | 88 |
| | **sub-total** | **141** | **126** | **267** |
| Integrated Care | Bugono HC | 47 | 42 | 89 |
| | Busowobi HC | 47 | 42 | 89 |
| | Baitambogwe HC | 48 | 42 | 90 |
| | **sub-total** | **142** | **126** | **268** |
| | **Uganda overall total** | **424** | **377** | **801** |
| **SOUTH AFRICA** | | | | |
| Facility care | Michael Mapongwana CHC | 141 | 140 | 281 |
| Integrated Care | Site B CHC | 140 | 145 | 285 |
| | **South Africa overall total** | **281** | **285** | **566** |

**Table 2. Baseline characteristics of participants.**

| Characteristic | Participants with T2D | | | | | Participants at high risk of T2D | | | | |
|---|---|---|---|---|---|---|---|---|---|---|
| | Uganda | | | South Africa | | Uganda | | | South Africa | |
| | Facility care (n = 141) | Integrated care (n = 142) | Usual care (n = 141) | Facility care (n = 141) | Integrated care (n = 140) | Facility care (n = 126) | Integrated care (n = 126) | Usual care (n = 125) | Facility care (n = 140) | Integrated care (n = 145) |
| Sex: | | | | | | | | | | |
| Females | 78 (55.3%) | 85 (60.0%) | 75 (53.2%) | 100 (70.9%) | 87 (61.1%) | 103 (81.8%) | 98 (77.8%) | 89 (71.2%) | 98 (70.0%) | 126 (86.9%) |
| | P = 0.440[§] | P = 0.513[Ω] | | P = 0.130[§] | | P = 0.433[§] | P = 0.135[Ω] | | P = 0.001[§] | |
| Age in years: | | | | | | | | | | |
| 30–45 | 38 (27.0%) | 31 (21.8%) | 29 (20.6%) | 32 (22.7%) | 48 (34.3%) | 38 (30.2%) | 43 (34.1%) | 34 (27.2%) | 31 (22.1%) | 51 (35.2%) |
| 46–60 | 79 (56.0%) | 84 (59.2%) | 80 (56.7%) | 85 (60.3%) | 66 (47.1%) | 62 (49.2%) | 54 (42.9%) | 61 (48.8%) | 79 (56.4%) | 66 (45.5%) |
| 60–75 | 24 (17.0%) | 27 (19.1%) | 32 (2.7%) | 24 (17.0%) | 26 (18.6%) | 26 (20.6%) | 29 (23.0%) | 30 (24.0%) | 30 (21.4%) | 28 (19.3) |
| | P = 0.596[§] | P = 0.606[Ω] | | P = 0.059[§] | | P = 0.599[§] | P = 0.735[Ω] | | P = 0.046[§] | |
| Mean ± SD | 52.5 ± 9.9 | 52.9 ± 9.6 | 52.6 ± 9.9 | 52.7 ± 9.3 | 50.1 ± 11.0 | 51.9 ± 10.9 | 50.6 ± 11.7 | 52.5 ± 11.2 | 53.5 ± 9.9 | 49.7 ± 10.6 |
| | P = 0.754[§] | P = 0.298[Ω] | | P = 0.029[§] | | P = 0.364[§] | P = 0.660 | | P = 0.002[§] | |
| Schooling attained: | | | | | | | | | | |
| None | 43 (30.5%) | 34 (23.9%) | 32 (22.7%) | 6 (4.3%) | 3 (2.1%) | 44 (34.9%) | 38 (30.2%) | 38 (30.4%) | 3 (2.1%) | 1 (0.7%) |
| Grade 1–7 | 69 (48.9%) | 78 (54.9%) | 73 (51.8%) | 41 (29.1%) | 28 (20.0%) | 58 (46.0%) | 65 (51.6%) | 71 (56.8%) | 38 (27.1%) | 28 (19.3%) |
| Above grade 7 | 29 (20.6%) | 30 (21.1%) | 35 (25.0%) | 94 (66.7%) | 109 (77.9%) | 24 (19.1%) | 23 (18.3%) | 16 (12.8%) | 99 (70.7%) | 116 (80.0%) |
| | P = 0.446[§] | P = 0.542[Ω] | | P = 0.075[§] | | P = 0.651[§] | P = 0.584[Ω] | | P = 0.179[§] | |
| Marital status | | | | | | | | | | |
| Never married | 2 (1.4%) | 0 (0.0%) | 3 (2.1%) | 27 (19.2%) | 26 (18.6) | 1 (0.8%) | 0 (0.0%) | 1 (0.8%) | 13 (9.3%) | 35 (24.1%) |
| Married/cohabiting | 105 (74.5%) | 103 (72.5%) | 103 (73.1%) | 82 (58.2%) | 81 (57.9%) | 85 (67.5%) | 90 (71.4%) | 82 (65.6%) | 78 (55.7%) | 69 (47.6%) |
| Separated/Divorced | 21 (14.9%) | 19 (13.4%) | 19 (13.5%) | 15 (10.6%) | 11 (7.9%) | 18 (14.3%) | 11 (8.7%) | 20 (16.0%) | 16 (11.4%) | 15 (10.3%) |
| Widowed | 13 (9.2%) | 20 (14.1%) | 16 (11.4%) | 17 (12.1%) | 22 (15.7%) | 22 (17.5%) | 25 (19.8%) | 21 (16.8%) | 33 (23.6%) | 26 (17.9%) |
| | P = 0.358[§] | P = 0.628[Ω] | | P = 0.734[§] | | P = 0.376[§] | P = 0.613[Ω] | | P = 0.009[§] | |
| Employment status | | | | | | | | | | |
| Employed in public or private sector | 9 (6.4%) | 13 (9.2%) | 14 (9.9%) | 42 (29.8%) | 56 (40.0%) | 5 (4.0%) | 9 (7.1%) | 8 (6.4%) | 65 (46.4%) | 31 (21.4%) |
| Self-employed | 24 (17.0%) | 23 (16.2%) | 24 (17.0%) | 11 (7.8%) | 8 (5.7%) | 13 (10.3%) | 21 (16.7%) | 21 (16.8%) | 10 (7.1%) | 7 (4.8%) |
| Unemployed | 9 (6.4%) | 3 (2.1%) | 9 (6.4%) | 65 (46.1%) | 61 (43.6%) | 8 (6.4%) | 5 (4.0%) | 5 (4.0%) | 47 (33.6%) | 86 (59.3%) |
| Peasant farmer | 92 (65.3%) | 93 (65.5%) | 87 (61.2%) | - | - | 98 (77.8%) | 87 (69.1%) | 88 (70.4%) | - | - |
| Retired | - | - | 3 (2.1%) | 18 (12.8%) | 13 (9.3%) | - | - | - | 16 (11.4%) | 16 (11.0%) |
| Other | 7 (5.0%) | 10 (7.0%) | 7 (5.0%) | 5 (3.6%) | 2 (1.4%) | 2 (1.6%) | 4 (3.2%) | 3 (2.4%) | 2 (1.4%) | 5 (3.5%) |
| | P = 0.381[§] | P = 0.660[Ω] | | P = 0.329[§] | | P = 0.285[§] | P = 0.638[Ω] | | P< 0.001[§] | |
| BMI[a] (kg/m$^2$) | | | | | | | | | | |
| < 25.0 | 67 (47.5%) | 57 (40.1%) | 67 (47.5%) | 12 (8.5%) | 14 (10.0%) | 62 (49.2%) | 71 (56.4%) | 66 (52.8%) | 16 (11.4%) | 16 (11.2%) |
| 25.0–29.9 | 49 (34.8%) | 53 (37.3%) | 47 (33.3%) | 22 (15.6%) | 44 (31.4%) | 38 (30.2%) | 31 (12.3%) | 38 (30.4%) | 27 (19.3%) | 36 (24.8%) |
| ≥ 30.0 | 25 (17.7%) | 32 (22.5%) | 27 (19.2%) | 107 (75.9%) | 82 (58.6%) | 26 (20.6%) | 24 (19.1%) | 21 (16.8%) | 97 (69.3%) | 93 (64.1%) |
| | P = 0.403[§] | P = 0.673[Ω] | | P = 0.004[§] | | P = 0.497[§] | P = 0.725[Ω] | | P = 0.527[§] | |
| Mean ± SD | 26.1 ± 7.4 | 26.6 ± 5.6 | 26.2 ± 8.1 | 36.1 ± 8.3 | 32.8 ± 7.0 | 26.8 ± 16.1 | 25.9 ± 6.6 | 25.2 ± 4.7 | 33.7 ± 7.0 | 34.1 ± 8.5 |
| | P = 0.533[§] | P = 0.343[Ω] | | P< 0.001[§] | | P = 0.570[§] | P = 0.630[Ω] | | P = 0.640[§] | |
| Current tobacco use n (%) | 3 (2.1%) | 5 (3.5%) | 5 (3.6%) | 21 (14.9%) | 11 (7.9%) | 1 (0.8%) | 2 (1.6%) | 3 (2.4%) | 19 (13.6%) | 26 (17.9%) |
| | P = 0.723[§] | P = 0.826[Ω] | | P = 0.090[§] | | P = 0.992[§] | P = 0.596[Ω] | | P = 0.313[§] | |
| Alcohol use, n (%) | | | | | | | | | | |

(*Continued*)

**Table 2.** (Continued)

| Characteristic | Participants with T2D | | | | | Participants at high risk of T2D | | | | |
| --- | --- | --- | --- | --- | --- | --- | --- | --- | --- | --- |
| | Uganda | | | South Africa | | Uganda | | | South Africa | |
| | Facility care (n = 141) | Integrated care (n = 142) | Usual care (n = 141) | Facility care (n = 141) | Integrated care (n = 140) | Facility care (n = 126) | Integrated care (n = 126) | Usual care (n = 125) | Facility care (n = 140) | Integrated care (n = 145) |
| 1–7 days/week | 10 (7.1%) | 13 (9.2%) | 11 (7.8%) | 9 (6.4%) | 18 (12.9%) | 11 (8.7%) | 18 (14.3%) | 11 (8.8%) | 13 (9.3%) | 37 (25.5%) |
| < 4 days/ mth | 2 (1.4%) | 5 (3.5%) | 3 (2.1%) | 18 (12.8%) | 17 (12.1%) | 1 (0.8%) | 4 (3.2%) | 1 (0.8%) | 16 (11.4%) | 21 (14.5%) |
| Never | 129 (91.5%) | 123 (86.6%) | 126 (89.4% | 114 (80.9%) | 104 (74.3%) | 114 (90.5%) | 104 (82.5%) | 113 (90.4%) | 111 (79.3%) | 87 (60.0%) |
| | P = 0.395$^{\S}$ | | P = 0.760 $^{\Omega}$ | P = 0.176$^{\S}$ | | P = 0.142$^{\S}$ | | P = 0.248 $^{\Omega}$ | P< 0.001$^{\S}$ | |
| Hypertensive$^{c}$, n (%) | 93 (64.0%) | 71 (50.0%) | 86 (61.0%) | 62 (44.0%) | 35 (25.0%) | 77 (61.1%) | 75 (59.5%) | 53 (42.4%) | 57 (40.7%) | 47 (32.4%) |
| | P = 0.007$^{\S}$ | | P = 0.020 $^{\Omega}$ | P = 0.001$^{\S}$ | | P = 0.797$^{\S}$ | | P = 0.004 $^{\Omega}$ | P = 0.146$^{\S}$ | |
| Equivalent of at least 150 mins of moderate physical activity/ wk? | | | | | | | | | | |
| Yes | 111 (78.7) | 122 (85.9%) | 118 (83.7%) | 111 (78.7%) | 99 (70.7%) | 109 (86.5%) | 116 (92.1%) | 111 (88.8%) | 114 (81.4%) | 118 (81.4%) |
| | P = 0.142$^{\S}$ | | P = 0.323 $^{\Omega}$ | P = 0.122$^{\S}$ | | P = 0.221$^{\S}$ | | P = 0.373 $^{\Omega}$ | P = 0.991$^{\S}$ | |
| HbA1c (%) | | | | | | | | | | |
| Mean ± SD | 9.6 ± 2.4 | 9.5 ± 2.3 | 9.5 ± 2.5 | 8.3 ± 2.1 | 8.9 ± 2.5 | 5.8 ± 0.73 | 5.7 ± 0.85 | 5.9 ± 0.8 | 5.9 ± 0.79 | 5.6 ± 0.83 |
| | P = 0.755$^{\S}$ | | P = 0.954 $^{\Omega}$ | P = 0.042$^{\S}$ | | P = 0.434$^{\S}$ | | P = 0.155 $^{\Omega}$ | P = 0.008$^{\S}$ | |

$^{*}$ Percentages are column percentages

$^{\S}$ p-value for comparison between the Integrated Care arm versus the Facility Care arm

$^{\Omega}$ p-value for comparison between the three study arms in Uganda

$^{c}$ Systolic blood pressure $\geq$ 140, and/or diastolic blood pressure $\geq$ 90, and/or on anti-hypertension treatment

differences between the study arms were observed with more female, younger and never married, and unemployed participants with higher level of alcohol use and lower mean HbA1c in the integrated care arm.

## Effectiveness of the trial interventions on glycemic control among participants with T2D

Among participants with T2D, bi-variable analysis showed a significantly higher percentage of glycemic control in Uganda at 29.2%, compared to 16.4% in South Africa (p< 0.001). We observed no significant differences in the crude percentages of participants with glycemic control between the study arms in either of the two countries. Adjusted analysis also revealed no significant differences in the estimated rates of glycemic control between any of the study arms in either of the two countries, except for the facility versus usual care arm in Uganda with an incidence rate ratio (IRR) of 0.71 [0.52–0.96] (Table 3).

## Effectiveness of the trial interventions on reduction in HbA1c levels among participants at high risk of T2D

Bi-variable analysis revealed a significantly higher percentage of participants at high risk for T2D achieved a reduction in HbA1c of at least 3 mmol/mol in Uganda at 36.6%, compared to 5.6% in South Africa (p< 0.001). In South Africa, we observed no statistically significant difference in the crude percentages between the two study arms. However in Uganda, the crude percentage of participants with reduction in HbA1c of at least 3 mmol/mol was significantly

**Table 3. Effectiveness of interventions on glycaemic control, and on reduction in HbA1c of at least 3 mmol/mol.**

| Country | Study arm | Glycaemic control among participants with T2D | | | | Reduction in HbA1c of at least 3 mmol/mol among participants at high risk of T2D | | | |
|---|---|---|---|---|---|---|---|---|---|
| | | number enrolled | Number with glycemic control (%) | Crude IRR [95% CI] | Adjusted IRR [95% CI] | number enrolled | Number with reduction in HbA1c $\geq$ 3 mmol/mol (%) | Crude IRR [95% CI] | Adjusted IRR [95% CI] |
| | Usual care | 141 | 46 (32.6%) | 1.0 | 1.0 | 125 | 22 (17.6%) | 1.0 | 1.0 |
| Uganda | Facility care | 141 | 42 (29.8%) | 0.67 [0.47–0.95] | 0.71 [0.52–0.96][a] | 126 | 47 (37.3%) | 0.92 [0.69–1.22] | 0.92 [0.70–1.22][a] |
| | Integrated care | 142 | 36 (25.4%) | 0.64 [0.44–0.94] | 0.66 [0.42–1.03][a] | 126 | 69 (54.8%) | 0.95 [0.71–1.27] | 0.97 [0.74–1.26][a] |
| | sub-total | 424 | 124 (29.2%) | | | 377 | 138 (36.6%) | | |
| | ICC = 0.028 | | | | | ICC = 0.043± | | | |
| South Africa | Facility care | 141 | 25 (17.7%) | 1.0 | 1.0 | 140 | 9 (6.4%) | 1.0 | 1.0 |
| | Integrated care | 140 | 21 (15.0%) | 1.01 [0.71–1.62] | 1.08 [0.71–1.62][b] | 145 | 7 (4.8%) | 1.00 [0.35–2.84] | 1.08 [0.38–3.10][b] |
| | sub-total | 281 | 46 (16.4%) | | | 285 | 16 (5.6%) | | |

ICC = Intra-Cluster Correlation coefficient

[¥]ICC in relationship to glycaemic control among participants with T2D in Uganda

[±]ICC in relationship to reduction in HbA1c > 3 mmol/mol among participants at high risk in Uganda

[a] Adjusted for clustering within health facility of enrolment; and for age group, sex, level of education, body mass index, and baseline HbA1C

[b] Adjusted for age-group, sex, BMI, level of education, and baseline HbA1c.

higher in the integrated care arm at 54.8%, followed by 37.3% in the facility care arm, and significantly lower in the usual care arm at 17.6% (p< 0.001). The crude percentage with reduction in HbA1c of at least 3 mmol/mol of 54.8% was significantly higher from the percentage with reduction in HbA1c of at least 3 mmol/mol in the facility care arm of 37.3% (p = 0.005).

Adjusted analysis revealed no significant differences in the estimated rates of reduction in HbA1c of at least 3 mmol/mol between the study arms in either of the two countries (Table 3).

## Effectiveness of the trial interventions on retention into care among participants with T2D

Overall among participants with T2D, bi-variable analysis revealed a significantly higher percentage of retention into care in Uganda at 75.0%, compared to 48.0% in South Africa (p< 0.001). In neither country did we observe a significant difference in the percentage of participants retained into care between the facility care arm and the integrated care arm. In Uganda, there was a significantly lower retention into care in the usual care arm at 58.9%, compared to the integrated care and facility care arms combined (83.0%) (p<0.001) (Table 4).

Adjusted analysis showed that in Uganda compared to the usual care arm, rates of retention into care were significantly higher in the facility care arm (IRR = 1.41 [1.08–1.84]), and the integrated care arm (IRR = 1.41 [1.09–1.83]). In South Africa, rates of retention into care were significantly higher in the integrated care arm compared to those in the facility care arm (IRR = 1.15 [1.10–1.19]) (Table 4).

## Effectiveness of the trial interventions on retention into care among participants at high risk of T2D

Among participants at high risk of T2D, a significantly higher percentage of retention into care was observed in Uganda at 57.6%, compared to 49.5% in South Africa (p = 0.039). No

**Table 4. Effectiveness of interventions on retention into care.**

| Country | Study arm | Retention into care among participants with T2D | | | | Retention into care among participants at high risk of T2D | | | |
|---|---|---|---|---|---|---|---|---|---|
| | | Number enrolled | Number retained into care (%) | Crude IRR [95% CI] | Adjusted IRR [95% CI] | Number enrolled | Number retained into care (%) | Crude IRR [95% CI] | Adjusted IRR [95% CI] |
| Uganda | Usual care | 141 | 83 (58.9%) | 1.0 | 1.0 | 125 | 31 (24.8%) | 1.0 | 1.0 |
| | Facility care | 141 | 116 (82.3%) | 1.40 [1.19–1.64] | 1.41 [1.08–1.84][g] | 126 | 77 (61.1%) | 2.46 [1.76–3.45] | 2.46 [1.33–4.53][m] |
| | Integrated care | 142 | 119 (83.8%) | 1.42 [1.22–1.66] | 1.41 [1.09–1.83][g] | 126 | 109 (86.5%) | 3.49 [2.55–4.77] | 3.52 [2.13–5.80][m] |
| | sub-total | 424 | 318 (75.0%) | | | 377 | 217 (57.6%) | | |
| | ICC$^{\Sigma}$ = 0.09306 (SE = 0.05142) | | | | | ICC$^{\Phi}$ = 0.33952 (SE = 0.11861) | | | |
| South Africa | Facility care | 141 | 73 (51.8%) | 1.0 | 1.0 | 140 | 68 (48.6%) | 1.0 | 1.0 |
| | Integrated care | 140 | 62 (44.3%) | 0.86 [0.67–1.09] | 1.15 [1.10–1.19][h] | 145 | 74 (51.0%) | 1.05 [0.83–1.33] | 1.02 [1.02–1.03][n] |
| | sub-total | 281 | 135 (48.0%) | | | 285 | 142 (49.8%) | | |
| | ICC$^{\Omega}$ = 0.00408 (SE = 0.01576) | | | | | ICC$^{\Psi}$ < 0.00001 (SE = 0.00994) | | | |

$^{\Sigma}$ ICC in relationship to retention into care among participants with T2D in Uganda

$^{\Phi}$ ICC in relationship to retention into care among participants at high risk in Uganda

$^{\Omega}$ ICC in relationship to retention into care among participants with T2D in South Africa

$^{\Psi}$ ICC in relationship to retention into care among participants at high risk in South Africa

[g] Adjusted for clustering within health facility of enrolment.

[h] Adjusted for clustering within health facility of enrolment; and for age group and baseline BMI.

[m] Adjusted for clustering within health facility of enrolment; and for sex.

[n] Adjusted for clustering within country site and health facility of enrolment; and for sex, baseline BMI, baseline hypertension status, and baseline HbA1c.

significant differences were observed in South Africa between the two study arms. However in Uganda, a significantly higher percentage of retention into care was observed in the integrated care arm at 86.5% compared to that in the facility care arm at 61.1% (p< 0.001). Moreover, a significantly lower percentage of retention into care was observed in the usual care arm at 24.8%, compared to the other two arms combined (p< 0.001) (Table 4).

Adjusted analysis showed that in Uganda, compared to the usual care arm, rates of retention into care were significantly higher in the facility care arm with an IRR = 2.46 [1.33–4.43], and in the integrated care arm with an IRR = 3.52 [2.13–5.80]. However there was no significant difference in rates of retention into care in the integrated care arm compared to the facility care arm.

In South Africa, rates of retention into care were significantly higher in the integrated care arm compared to the facility care arm, with an IRR = 1.02 [1.02–1,03) (Table 4).

**Effectiveness of the trial interventions on Incident T2D among participants at high risk.** A total of 328 participants at high risk for T2D had an HbA1c reading of less than 6.5% at baseline. Of these, 15 (4.6%) progressed to T2D; of which 9 out of 217 (4.1%) were in Uganda, and 6 out of 142 (4.2%) were in South Africa (p = 0.971). We found no significant difference in the percentage of participants at high risk that progressed to T2D between the study arms, neither in Uganda, nor in South Africa. Because of the small number of incident T2D cases observed in the study, adjusted analysis comparing incident rates of T2D between the study arms was not possible.

**Intra-Cluster Correlation coefficients.** The intra-cluster correlation coefficients, in respect to each study outcome of interest are presented in Tables 3 and 4.

## Discussion

We found significant improvements in retention into care comparing improved facility care to current usual care, and an added value of integrated care that included peer support in both countries. Overall, the effect size estimates of integrated care on retention into care were larger in Uganda, especially among participants at high risk of T2D.

The significant effect of the interventions on retention into care implies that a considerably less proportion of participants was lost to follow-up in the intervention arms of the study, that is, in South Africa the integrated care arm; and in Uganda the facility care and integrated care arms. This effect was relatively high in some study arms compared to others, resulting in highly unbalanced data between control (i.e. usual care) and intervention arms (i.e. facility only or integrated care). Because of this considerable and unbalanced loss to follow-up, the validity of our findings with regards to the effect the study intervention on the primary outcomes of glycaemic control and reduction in HbA1c might be questionable, despite having used multiple imputations [28] to address the missing data during analysis. Sensitivity analysis comparing the results of the multiple imputation analysis with those of intention-to-treat in which all cases lost to follow-up were set 'not having the outcome of interest' [20] (see S1 Table), reveals strong differences between both approaches which may further question the validity of our results. The unexpectedly high number of missingness also reduced the power of our study which may explain why we did not find significant differences between the study arms for the primary outcomes. We conclude that in pragmatic implementation trials such as the present study with the objective to enhance retention into care and glycemic control, may result in unbalanced missingness, hence compromising further assessment of the effectiveness.

Although there is evidence for community-based peer support interventions in high-income countries, research from LMICs have shown inconsistent associations with improvements in clinical, behavioral, and psychological outcomes [6]. In our study, all positive findings were similar for both the integrated care arm and the facility care arm in Uganda, suggesting a stronger role for facility care. Still in Uganda, whereas community-based peer support strategies added no value for participants with T2D, participants at high risk benefited from integrated care in terms of retention into care. Put together, our findings point to the need for an adequately functioning primary care system for T2D management and a role of the community for T2D prevention.

In the only previous study on peer support for diabetes self-management we were able to find in the Uganda context [29], Baumann et al (2015) conducted a small quasi-experimental study and their findings indicated improvements in HbA1c, diastolic blood pressure, and dietary patterns. In our study, the intervention did not show a significant effect on reduction in HbA1c of at least 3 mmol/nol, but as mentioned before, the unbalanced loss-to-follow-up compromised valid analysis of the data. It is notable, however, that minimal changes at the facility level such as ensuring no medication stock-outs, availability of clinical monitoring equipment, etc. in Uganda, led to significant improvements in retention into care; especially among participants at high risk of T2D. Stock-outs and poor availability of essential medication and diagnostic tests are known barriers to diabetes and non-communicable disease management and prevention in Sub-Saharan Africa [30, 31].

Further in Uganda where the standard of usual care is low for patients with T2D, and participants at high risk remain largely unidentified and untreated [32], our study was able to significantly improve retention into care in both participant groups, and in both intervention arms. However, people without a T2D diagnosis but with a high-risk diagnosis are a new patient group for health facilities and it seems community involvement was able to further improve their retention into care and HbA1c outcomes. Taken together, the findings from

Uganda call for further research on potential mediating factors related to self-management behaviors as well as implementation outcomes.

The non-significant findings in South Africa in regards to reduction in HbA1c were similar to an earlier peer support intervention in the same context [33] which showed negligible effect in terms of primary and secondary outcomes. In that study, most intervention participants had not attended a single intervention session, and a qualitative process evaluation [34] revealed that intervention fidelity was also constrained. Despite continued efforts to monitor intervention fidelity in our study, it is probable that failure to achieve a significant difference in HbA1c in South Africa could be attributed to inadequate intervention implementation rather than intervention efficacy. Findings of an extensive process evaluation are currently being analyzed and already point to several issues, such as a low absorptive capacity among Community Health Workers (CHWs), insufficient supervision of CHWs and rapid staff turn-over and organizational change at the level of the implementing partner; security concerns, and low engagement of study participants which proved difficult without incentivization (e.g. FPG and BP measurement).

In their systematic review on peer support interventions for diabetes self-management Werfalli et al [6] pointed at several reasons for inconsistent findings: low quality of studies, implementation issues, and lack of sufficient (contextualized) underlying theory to build the intervention. As discussed above, our study faced many challenges related to implementation especially in South Africa. Another challenge was related to selection of health care facilities as trial clusters. In Uganda, the study area was large enough to avoid contamination between trial arms and to have enough clusters for successful randomization, which resulted in minor differences only between the study arms at baseline. In South Africa, where many other studies were conducted simultaneously in the same geographically restricted area, only two facilities could be selected for feasibility reasons, with large differences in demographic catchment areas as reflected in the baseline differences between both study arms. Among participants with T2D and those at high risk, the integrated care arm had significantly younger participants with a better risk factor profile. These differences imply that participants in the integrated care arm might have been less likely to adhere to the intervention due to e.g., work and family obligations, and less likely to show benefits of the intervention due to their risk profile.

In settings like the Uganda rural study setting where the current standard of care for people at risk of or with T2D is still poor, basic improvements in facility care improves retention into care. In this context we also see added value of community-based peer support to provide integrated care. We therefore recommend implementation of such measures, as minimal resources are needed. However, in South Africa, a middle-income urban setting with higher facility standard of care and highly mobile populations, we could not demonstrate added value of integrated interventions beyond facility care. Here, more studies are called for to identify strategies that can further increase retention and glycemic control.

## Strengths and limitations

The SMART2D pragmatic cluster randomized trial evaluated the added benefit of community intervention strategies on type 2 diabetes outcomes, beyond optimized health facility strategies in Uganda and South Africa, exemplifying a low- and middle-income setting that are facing challenges in tackling the type 2 diabetes burden. The study was designed with a solid evidence-base including, for example, the Self-Determination Theory (SDT), and utilized evidence-based and contextualized strategies contextualized based upon formative research and collaboration with local and sub-national stakeholders, to ensure relevance, acceptability and feasibility for scale-up. The study was able to demonstrate that when minimum level quality

care is lacking, even minimal improvements in the facility care like training nurses to follow-up patients and encourage participants to keep appointments; can provide significant improvements in retention into care and, potentially, in diabetes management outcomes. However, since the interventions were implemented as a package of intervention elements, we were unable to identify specific intervention elements that contributed towards improved T2D prevention and diabetes management outcomes.

A major strength of our study was its theoretical underpinning. Many intervention studies either lack any explicit theory of change or they utilize a theory that has not been created or tested in a LMIC context. Our formative research provided evidence for SDT with regards to physical activity in the Uganda site and healthy eating and physical activity in the South Africa site [35–37]. Furthermore, SDT formed an essential part of the SMART2D self-management framework developed for this study [15]. Other strengths include the comprehensive situational analysis in all participating country sites [10, 15, 35], participatory intervention development including research teams and key stakeholders [11], and a comprehensive process evaluation aiming for a deeper understanding of the effect of the intervention components through the analyses of secondary outcomes as potential mediators of change, as well as enablers and barriers for implementation. Together, these analyses can further explain which, why and how specific components were implemented and this may explain change in secondary and main outcomes.

Our study had some limitations that are important to point out. First, due to several ongoing studies in the South African study site, selection of clusters was limited and resulted in several baseline differences between the study arms that might have influenced the findings. Furthermore, ours was a complex health intervention and each intervention strategy had one or more intervention elements as described in the trial protocol publication [13]. Intervention elements differed slightly between both countries because of contextual adaptation. Complex health interventions are not usually generalizable except for their main ingredients, as implementation in different settings should always take context into account.

## Supporting information

**S1 Table. Effectiveness of interventions on glycaemic control, and on reduction in HbA1c of at least 3 mmol/mol (using intention-to-treat analysis strategy).**
(DOCX)

**S1 Text. The SMART2D trial protocol.**
(DOCX)

**S2 Text. Completed consort checklist with extension for cluster trials.**
(DOCX)

**S1 Data. De-identified dataset.**
(DTA)

## Acknowledgments

We are grateful to the institutional support of the country site institutions to the SMART2D consortium. The SMART2D consortium includes the following six partner institutions: Makerere University, School of Public Health, Uganda; the University of Western Cape, School of Public Health, South Africa; Karolinska Institutet and Uppsala University, Sweden; Institute of Tropical Medicine, Belgium; and Collaborative Care Systems Finland. We acknowledge the contribution of the other SMART2D Consortium members that include:

Göran Tomson, Carl Johan Sundberg, Helle Mölsted Alvesson, David Sanders (RIP), Barbara Kirunda, Anthony Muyingo, Gloria Naggayi, Ronald Kusolo, and Edward Ikona. We also acknowledge the study participants for volunteering to participate in the trial, the contribution of the participating health centers and their staff in Uganda and South Africa including The Caring Network Community Health Workers and management, Diabetes SA, Katherine Murphy and Buyelwa Majikela-Dlangamanga from the Chronic Disease Initiative for Africa, Tshilidzi Manuga and Sunday Onagbiye and the entire field worker team from South Africa.

## Author Contributions

**Conceptualization:** David Guwatudde, Pilvikki Absetz, Josefien Olmen Van, Elizabeth Ekirapa Kiracho, Juliet Kiguli, Thandi Puoane, Claes-Goran Ostenson, Stefan Peterson, Meena Daivadanam.

**Data curation:** David Guwatudde, Josefien Olmen Van, Roy William Mayega, Mariam Hassen, Juliet Kiguli, Thandi Puoane, Meena Daivadanam.

**Formal analysis:** David Guwatudde.

**Funding acquisition:** Stefan Peterson, Meena Daivadanam.

**Investigation:** David Guwatudde, Peter Delobelle, Pilvikki Absetz, Josefien Olmen Van, Roy William Mayega, Francis Xavier Kasujja, Jeroen De Man, Mariam Hassen, Juliet Kiguli, Thandi Puoane, Claes-Goran Ostenson, Stefan Peterson, Meena Daivadanam.

**Methodology:** David Guwatudde, Peter Delobelle, Pilvikki Absetz, Josefien Olmen Van, Roy William Mayega, Francis Xavier Kasujja, Jeroen De Man, Elizabeth Ekirapa Kiracho, Juliet Kiguli, Thandi Puoane, Claes-Goran Ostenson, Stefan Peterson, Meena Daivadanam.

**Project administration:** David Guwatudde, Peter Delobelle, Pilvikki Absetz, Francis Xavier Kasujja, Thandi Puoane, Meena Daivadanam.

**Resources:** Stefan Peterson, Meena Daivadanam.

**Supervision:** David Guwatudde, Peter Delobelle, Josefien Olmen Van, Roy William Mayega, Francis Xavier Kasujja, Mariam Hassen, Thandi Puoane, Meena Daivadanam.

**Validation:** David Guwatudde, Peter Delobelle, Meena Daivadanam.

**Visualization:** David Guwatudde, Peter Delobelle, Pilvikki Absetz, Meena Daivadanam.

**Writing – original draft:** David Guwatudde.

**Writing – review & editing:** David Guwatudde, Peter Delobelle, Pilvikki Absetz, Josefien Olmen Van, Roy William Mayega, Francis Xavier Kasujja, Jeroen De Man, Mariam Hassen, Elizabeth Ekirapa Kiracho, Juliet Kiguli, Thandi Puoane, Claes-Goran Ostenson, Stefan Peterson, Meena Daivadanam.

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
