## [Decision Letter · Decision Letter 0]

25 Jan 2022

PGPH-D-21-00724

Prevention and management of type 2 diabetes mellitus in Uganda and South Africa: Findings from the SMART2D pragmatic implementation trial.

Dear Dr. Guwatudde,

Thank you for submitting your manuscript to PLOS Global Public Health. After careful consideration, we feel that it has merit but does not fully meet PLOS Global Public Health’s publication criteria as it currently stands. Therefore, we invite you to submit a revised version of the manuscript that addresses the points raised during the review process.

We look forward to receiving your revised manuscript.

Kind regards,

Paolo Angelo Cortesi, PhD

Academic Editor

Journal Requirements:

1. Please ensure that the funders and grant numbers match between the Financial Disclosure field and the Funding Information tab in your submission form. Note that the funders must be provided in the same order in both places as well. 

Funding Information:

Funder Name: H2020 European Research Council

Grant Number: 643692

Grant Recipient: Meena Daivadanam

Funder Name: Agence Nationale de Recherches sur le Sida et les Hépatites Virales

Grant Number: HS 343

Grant Recipient: Prof David Guwatudde

Financial Disclosure:

This study was part of the SMART2D project funded by the European Commission's Horizon2020 Health Coordination Activities (Grant Agreement No 643692) under call “HCO-05-2014: Global Alliance for Chronic Diseases: prevention and treatment of type 2 diabetes”. The Uganda site was co-funded by the Sweden International Development Cooperation Agency (Sida) capacity-building grant to Makerere University 2015-2010, Project number HS 343. The contents of this article are solely the responsibility of the authors and do not reflect the views of the funders of the SMART2D Project.

2. Please update your Competing Interests statement. If you have no competing interests to declare, please state: “The authors have declared that no competing interests exist.”

3. In the online submission form, you indicated that

"The original study data is available from the corresponding author, and can be made available upon request and approval of co-investigators on:

David Guwatudde,

Department of Epidemiology and Biostatistics,

School of Public Health,

Makerere University College of Health Sciences,

Kampala, Uganda.

Email: dguwatudde@musph.ac.ug".

4. We have noticed that you have uploaded supporting information but you have not included a list of legends. Please add a full list of legends for all supporting information files (including figures, table and data files) after the references list.

Additional Editor Comments (if provided):

Reviewer #1: General Comment

Diabetes Mellitus is a multisystem disease that is now a major public health concern worldwide. This is an interesting study and exciting to review. The manuscript is generally well written and structured, and will be a good read for researchers, clinicians, and the health promotion and disease prevention communities. Specific comments have been listed below.

Specific comments

Consider including a brief introduction or background in the abstract

Line 39: The design is not reproducible. clarity with precision will be required.

Line 52: Be specific, indicate p-values

Line 54: indicate p-values

Line 65: Full stop to be placed after the reference citation, not before: example - "….especially in low- and middle-income counties (LMICs)"[1]. Consider this in entire text.

Line 68, comma after "2019"

Lines 74-76, "..18.1% were treated but uncontrolled, and only 19.4% were treated and controlled; suggesting that 80.6% of the population with diabetes had unmet need for care". Consult your reference and reword for simplicity.

Clinical definition of Type 2 DM should be stated in your introduction or outcome section

Line 127, comma after criteria

Line 187: How long was the intervention?

Line 223: since this is a pre-post intervention study, was there a consideration to evaluate the possible effect of the intervention using the dependent/paired t- test?

Line 253: check spelling - baseline

Table 1: The break-off to new page must capture the headings as in previous page.

Line 323: Significance should be relative to p-value and not percentage in each country. It is vital to clearly indicate how the post intervention characteristics differed from the baseline characteristic in each country/study area using an appropriate statistical test.

Table 3: "0.00128¥" Should this be here in this column or the next column? If this is the p-value, check if this contradicts the statement in lines 324-325.

Table 3": write in full. Avoid using "#" as headings

Table 3: place the p-values for the ICC at the appropriate columns dedicated for p-values

Table 6: regarding the Adjusted IRR, were there measures in place to control confounding factors?

Table 6: Any explanation why there is no 'usual care' in South Africa?

Line 353, hba1c to be written appropriately - HbA1c

Line 399: Was there any confounding factor(s) that may have affected the percentage outcome based on the crude IRR and adjusted models?

Lines 406-407: "….rates of retention into were significantly higher in…" Seems incomplete; possible omission. Read again and revise accordingly.

Line 411: Cited table in text should be in circle brackets.

Reviewer #2: This is an important addition to literature. It is novel and topical. I strongly support its publication. However, in its present form, it is really difficult to understand the study design, who the patients were (e.g. how selected) and therefore what do the findings mean.

The abstract lacks details and is impossible to understand. What are the 3 arms, what is a high risk patient and what is the difference between glycaemic control and HbA1c control? I don't understand the rate ratios in the abstract.

Even in the methods, it is unclear what are the different arms? What is the difference between facility care and usual care. The latter is not described. Perhaps the authors should consider a Table / chart to describe these side by side so that the reader can see the difference. Also think of better names for these arms that bring great clarity.

In the outcomes section, please explain what does 3 mmol/mol mean clinically.

The study design is not clear. Is this a randomised trial, with randomisation done by clusters? And how were participants selected from the pool of people with diabetes or at high risk.

The results section is very difficult to follow with too many comparisons and too many outcomes. The authors really must focus on their primary questions, which I guess are the effects of the different models of care. I can't work out whether the analyses in T4 tally with those in T3. These 2 tables should really be combined into one table.

Why is there such a powerful effect of integrated care among high-risk but not among people with T2D on glycaemic indicators. Does seem a bit odd?

Likewise, T5 and T6 ought to be combined into one table.
---

## [Decision Letter · Decision Letter 1]

6 Apr 2022

Prevention and management of type 2 diabetes mellitus in Uganda and South Africa: Findings from the SMART2D pragmatic implementation trial.

PGPH-D-21-00724R1

Dear Dr. Guwatudde,

We are pleased to inform you that your manuscript 'Prevention and management of type 2 diabetes mellitus in Uganda and South Africa: Findings from the SMART2D pragmatic implementation trial.' has been provisionally accepted for publication in PLOS Global Public Health.

Best regards,

Paolo Angelo Cortesi, PhD

Academic Editor

Reviewer #1: The authors have satisfactorily addressed my comments.

Reviwer #2: I think the authors have made good changes and I now recommend this paper be published. So no further comments from me.